# Tailoring Upconversion and Morphology of Yb/Eu Doped Y_2_O_3_ Nanostructures by Acid Composition Mediation

**DOI:** 10.3390/nano9020234

**Published:** 2019-02-09

**Authors:** Daniela Nunes, Ana Pimentel, Mariana Matias, Tomás Freire, A. Araújo, Filipe Silva, Patrícia Gaspar, Silvia Garcia, Patrícia A. Carvalho, Elvira Fortunato, Rodrigo Martins

**Affiliations:** 1i3N/CENIMAT, Department of Materials Science, Faculty of Sciences and Technology, Universidade NOVA de Lisboa and CEMOP/UNINOVA, Campus de Caparica, 2829-516 Caparica, Portugal; ms.matias@campus.fct.unl.pt (M.M.); tm.freire@campus.fct.unl.pt (T.F.); emf@fct.unl.pt (E.F.); 2INCM, Imprensa Nacional-Casa da Moeda, Av. António José de Almeida, 1000-042 Lisboa, Portugal; Andreia.Cardoso@incm.pt (A.A.); filipe.silva@incm.pt (F.S.); patricia.gaspar@incm.pt (P.G.) Silvia.Garcia@incm.pt (S.G.); 3SINTEF Materials Physics, Forskningvein 1, 0373 Oslo, Norway; patricia.carvalho@sintef.no

**Keywords:** microwave synthesis, oxide dissociation, doping, rare earth ions, upconversion

## Abstract

The present study reports the production of upconverter nanostructures composed by a yttrium oxide host matrix co-doped with ytterbium and europium, i.e., Y_2_O_3_:Yb^3+^/Eu^3+^. These nanostructures were formed through the dissociation of yttrium, ytterbium and europium oxides using acetic, hydrochloric and nitric acids, followed by a fast hydrothermal method assisted by microwave irradiation and subsequent calcination process. Structural characterization has been carried out by X-ray diffraction (XRD), scanning transmission electron microscopy (STEM) and scanning electron microscopy (SEM) both coupled with energy dispersive X-ray spectroscopy (EDS). The acid used for dissociation of the primary oxides played a crucial role on the morphology of the nanostructures. The acetic-based nanostructures resulted in nanosheets in the micrometer range, with thickness of around 50 nm, while hydrochloric and nitric resulted in sphere-shaped nanostructures. The produced nanostructures revealed a homogeneous distribution of the doping elements. The thermal behaviour of the materials has been investigated with in situ X-Ray diffraction and differential scanning calorimetry (DSC) experiments. Moreover, the optical band gaps of all materials were determined from diffuse reflectance spectroscopy, and their photoluminescence behaviour has been accessed showing significant differences depending on the acid used, which can directly influence their upconversion performance.

## 1. Introduction

Upconverter materials are characterized by having the ability to emit photons with higher energy than the photons that were absorbed [1]. The most studied upconverters use rare earth doped luminescent materials. Rare earth elements belong to the lanthanides group in the periodic table and typically are trivalent ions (Ln^3+^) in which the 4f inner shell is filled up to 14 electrons, presenting a 4f^n^5s^2^5p^6^ electron configuration (with n = 0–14) [1]. The partial filled 4f shell is the responsible for the optical and magnetic properties presented by this type of materials, and for the generation of radiation of different wavelengths, depending on the dopants used [2]. The most common used lanthanides elements are: Eu^2+^ and Ce^3+^ (broad band emitting due to the 5d → 4f transition) and Eu^3+^, Tb^3+^, Gd^3+^, Yb^3+^, Dy^3+^, Sm^3+^, Tm^3+^, Er^3+^, and Nd^3+^ (narrow band emitting due to transitions between 4f levels) [3]. Most of them present visible emissions, and some are described in Table 1 [4,5,6].

In the past few years, these rare earth oxides have attracted an increasing attention due to their possible use in several applications such as luminescence devices [7,8], color displays [9,10], optical detectors [11,12], telecommunications [5], upconverter lasers and photonics [13,14,15], catalysis [16], biomedicine [17,18] and solar cells [1,19,20], mainly due to their chemical, electronic, and optical properties resulting from the 4f electronic shells [21]. These properties are greatly dependent on the crystal structure, morphology and also on the chemical composition (type of dopants used), which are very sensitive to the bonding states of rare earth ions [21].

Yttrium oxide, Y_2_O_3_, is a rare earth oxide with a relatively high band gap value, around 5.8 eV [22], low phonon frequency and ionic radii [2]. Y_2_O_3_ is considered to be one of the greatest luminescent oxide phosphor materials available, mainly due to its high luminescence efficiency, chemical, thermal and photochemical stabilities but also due to its high color purity [23,24,25], presenting a large dielectric constant and good infrared transmission [22]. 

Y_2_O_3_ is a host material (matrix) and its wide band gap makes it attractive for several optical applications in the visible and UV ranges. Doping Y_2_O_3_ with other rare earth elements can bring some advantages, since they can emit within its optical window and not suffer quenching effects [26].

Ytterbium, Yb^3+^ is often used as a sensitizer due to its higher absorption cross-section that corresponds to 980 nm excitation, which will excite other elements like europium, Eu^3+^ [2]. Europium, on the other hand, is a trivalent material that contains ions from the rare earth and from the transition-metal series [27]. Doping Y_2_O_3_ with Eu^3+^ is considered to be one of the best red oxide phosphors due to their chemical stabilities and luminescence efficiencies [21,23]. The visible emission spectrum of Eu^3+^ is composed by ^5^D_0_ → ^7^F_2_, ^5^D_0_ → ^7^F_1_ and ^5^D_0_ → ^7^F_0_ [4]. The ratio between transitions ^5^D_0_ → ^7^F_2_ and ^5^D_0_ → ^7^F_1_ intensities, is an indicative of the resulting emission color. The higher the ratio, the more intense into the orange color will be the luminescence [4].

Y_2_O_3_ nanoparticles doped with Yb^3+^ and Eu^3+^ ions are expected to possess the ability to absorb near-infrared radiations (NIR) and upconvert it into visible radiation [28]. Yb^3+^ will work as a sensitizer (possessing one energy level around 980 nm, with a lifetime of ~2 ms, ideal for infrared to visible upconvertion), that will transfer the NIR energy to the activator Eu^3+^, with an emission wavelength shorter than the NIR excitation [1,28].

In terms of nanostructure shape, the synthesis of nanosized spherical phosphors particles brings the advantage of enhanced brightness and high resolution in upconversion applications [29], but also high packing densities and low light scattering. Therefore, it is suggested that the ideal morphology for a phosphor nanoparticle is a perfect spherical shape, with a narrow size distribution [29]. Several methods have already been reported for the synthesis of phosphors Y_2_O_3_ nanostructures aiming to obtain spherical shaped particles with homogeneous sizes and distributions, such as spray pyrolysis [23], sol-gel [30,31], hydrothermal/solvothermal methods [29], solution combustion [32,33], precipitation [34,35,36] and template method. Microwave assisted synthesis has also been reported for the production of Y_2_O_3_ materials [24,37,38], having numerous advantages, such as its celerity, reproducibility and cost-efficiency, in such a way that in the last few years it has been widely used on the synthesis of this oxide and several other types of oxides [39,40,41]. 

The present work focuses on the dissociation of primary oxides using different acids and further microwave synthesis of nanostructures based on Y_2_O_3_ doped with Yb and Eu. This synthesis route required minimal time to effectively dope the nanostructured matrix (15 min), which makes this a fast approach to produce such materials. The obtained nanostructures were fully characterized by X-ray diffraction (XRD), scanning transmission electron microscopy (STEM) and scanning electron microscopy (SEM), differential scanning calorimetry (DSC), Ultra Violet- Visible – Near Infrared (UV-VIS-NIR) spectroscopy, and photoluminescence.

## 2. Synthesis and Characterization of Yb/Eu Doped Y_2_O_3_ Nanostructures

### 2.1. Materials

The influence of different acids on the production of upconverter nanostructures under microwave irradiation was studied. For all the materials produced, the microwave synthesis was repeated at least 3 times for each set of samples produced, showing that this technique is highly reproducible, and presenting, within each set, the same type of structure, morphology and composition.

Regardless of the chosen acid, the basis of this synthesis is composed by a host matrix of yttrium oxide (Y_2_O_3_, CAS: 1314-36-9, purity 99.99%, with a particle size < 50 nm, from Sigma Aldrich, (Sigma-Aldrich Chemie Gmbh, Munich, Germany), and the dopants coming from the ytterbium oxide (Yb_2_O_3_, CAS: 1314-37-0, purity 99.7%, with a particle size < 100 nm, from Sigma Aldrich) and europium oxide (Eu_2_O_3_, CAS: 1308-96-9, purity 99.5%, with a particle size < 150 nm, from Sigma Aldrich), here named as the primary oxides. Several different acids have been used to dissolve the primary oxides and convert them to the form of nitrates: acetic acid (CH_3_COOH, CAS: 64-19-7, from Fisher Chemical (Fisher Scientific GmbH, Schwerte, Germany), hydrochloric acid (HCl, CAS: 7647-01-0, 37%, from Fisher Chemical) and nitric acid (HNO_3_ CAS: 7697-37-2, 69%, from Labkem(Blanc-Labo SA, Lonay, Switzerland). All the initial chemicals were used without further purification.

### 2.2. Yb/Eu Doped Y_2_O_3_ Synthesis

Three solutions were prepared, each of them with a different acid, *i.e.*, acetic, hydrochloric and nitric acids. In this way, it became possible to study the influence of different acids compositions in the final materials’ outcomes, resorting to a hydrothermal synthesis assisted by microwave irradiation.

In a typical synthesis, 0.05 M of Y_2_O_3_, 0.01 M of YbO_3_ and 0.0075 M of Eu_2_O_3_ were diluted in 5 mL of acid (CH_3_COOH, HCl or HNO_3_), resulting in the formation of a colorless solution. The solution was kept under stirring on a hot plate at 70 °C in ambient atmosphere, until complete evaporation. A white powder was then obtained.

Afterwards, 40 mL of H_2_O and a most common used catalyst [2,42,43,44,45] was added to this powder and left stir for 10 minutes until complete dissociation. The solution was then placed in a Pyrex vessel and placed into a microwave heating system, Discover SP, from CEM (CEM Corporation, Matthews, NC, USA), fixing the maximum power up to 100 W with a temperature of 130 °C and 15 min synthesis time.

The obtained powders were then washed with deionized water and isopropyl alcohol. Between each washing process, the powder was centrifuged at 3000 rpm for 3 min. The annealing of the obtained nanostructures was carried out in a Nabertherm furnace (Nabertherm GmbH, Lilienthal, Germany), at a temperature of 700 °C for 6 h, under atmospheric conditions, with a heating ramp of 1 h. Figure 1 shows a schematic of Yb/Eu doped Y_2_O_3_ nanostructures production steps.

### 2.3. Yb/Eu Doped Y_2_O_3_ Characterization

All the characterization measurements were carried out in the powder form. The crystallinity and phase purity of as-synthesized and annealed materials were characterized by X-ray diffraction, using a PANalytical’s X’Pert PRO MRD X-ray diffractometer (PANalytical B.V., Almero, The Netherlands), with a monochromatic CuKα, λ = 1.540598 Å, radiation source. The XRD measurements were carried out from 25° to 60°, with a scanning step size of 0.016°. The in situ diffractograms were collected in the same *2*θ range and at temperatures in steps of 100 °C, from 300 °C to 1000 °C. The material was kept at each temperature step for at least half an hour to allow the stabilization and 5 consecutive scans were collected for inspection of structural modifications during this time. The temperature was increased with a rate of ~1.7 °C min^−1^.

Differential Scanning Calorimetry measurements were carried out to the obtained powder, before annealing. It was used a simultaneous thermogravimetric analyzer, TGA-DSC-STA 449 F3 Jupiter, from Netzsch (Netzsch-Geratebau GmnH, Selb, Germany). Approximately 5 mg of powder was loaded into an open PtRh crucible and heated from room temperature to 1100 °C, with a heating rate of 10 °C min^−1^ under atmospheric conditions.

The morphology of all materials was analyzed by using a Scanning Electron Microscope (SEM) Carl Zeiss AURIGA CrossBeam Workstation instrument (Carl Zeiss Microscopy GmbH, Oberkochen, Germany), equipped with an Oxford energy dispersive X-ray spectrometer (EDS, Oxford Instruments Nanoanalysis, High Wycombe, UK). 

Scanning Transmission Electron Microscopy (STEM) analyses of the nanostructures were carried out at 300 kV with a FEI Titan G2 60-300 instrument (Thermo Fisher Scientific, Hillsboro, OR, USA) equipped with a DCOR probe Cs-aberration corrector and a Super-X Bruker energy dispersive spectrometer with 4 silicon drift detectors. The size of 40 individual nanocrystals was determined through STEM images using ImageJ software (version 1.52k) [46].

For band gap determination, diffuse reflectance measurements were carried out at room temperature, using a Perkin Elmer lambda 950 UV/VIS/NIR spectrophotometer (Perkin Elmer, Inc., Waltham, MA, USA), with a diffuse light detection module (150 mm diameter integrating sphere, internally coated with Spectralon). The calibration of the system was performed by using a standard reflector sample (reflectance, R, of 1.00 from Spectralon disk). The reflectance (R) spectra was acquired in the range between 175 and 1200 nm.

Photoluminescence were also examined with a 976 nm laser (Avantes NIR light, Iso-Tech power source 1 W, 5 nm resolution) between 500 and 750 nm. All measurements were carried out under identical excitation/detection conditions, and for properly comparison of the PL intensities, the powder was weighted and pressed under the same pressure to obtain the same powder density.

## 3. Results and Discussion

The reactions between the different acids and the host matrix oxide (Y_2_O_3_) can be described by the following Equations (from 1 to 3) [47,48]:
(1)$$\begin{array}{cc} \mathrm{Acetic}\;\mathrm{acid}: & Y_{2}O_{3}+2CH_{3}COOH+H_{2}O\to {\left(Y\left(CH_{3}COO\right){\left(OH\right)}_{2}\right)}_{2} \end{array}$$
(2)$$\begin{array}{cc} \mathrm{Hydrochloric}\;\mathrm{acid}: & Y_{2}O_{3}+3HCl\to YCl_{3}+Y{\left(OH\right)}_{3} \end{array}$$
(3)$$\begin{array}{cc} \mathrm{Nitric}\;\mathrm{acid}: & Y_{2}O_{3}+6HNO_{3}\to 2Y{\left(NO_{3}\right)}_{3}+3H_{2}O \end{array}$$

By mixing the catalyst to the yttrium ions obtained from the previous reactions, yttrium hydroxycarbonate is formed (Equation (4)). After microwave irradiation, it is observed the formation of yttrium oxycarbonate (Equation (5)). The powder formed is thus subjected to the final calcination step, in which the yttrium oxide nanostructures are obtained (Equation (6)) [43,45]:(4)$$2Y^{3+}+7H_{2}O+3CH_{4}N_{2}O\to 2YOHCO_{3}+4NH_{4}^{+}+CO_{2}$$
(5)$$2YOHCO_{3}\overset{Synthesis}{\to }Y_{2}O_{2}CO_{3}+H_{2}O+CO_{2}$$
(6)$$Y_{2}O_{2}CO_{3}\overset{annealing}{\to }Y_{2}O_{3}+CO_{2}$$

### 3.1. Structural Characterization of Yb/Eu Doped Y_2_O_3_ Nanostructures

Figure 2 shows the XRD diffractograms of the nanostructures obtained by microwave synthesis, when using different acids for the dissociations of the primary oxides, and before any heat treatment. It is possible to observe that the as-synthesized nanostructures are amorphous when using hydrochloric and nitric acid. Regarding the use of acetic acid, the synthesized nanostructures present some degree of crystallinity.

To find out the temperature at which there was an amorphous to crystalline phase transformation, *in situ* XRD analyses were carried out in the temperature range from 300 °C to 1000 °C. From the annealing process, in 100 °C steps, it is expected to complete the formation of crystalline Yb/Eu doped Y_2_O_3_ nanostructures from the intermediate products and sesquioxide phases of Y_2−x_O_3−x_ [23], as the temperature is raised. On Figure 3, it is possible to observe the XRD diffractograms evolution with the increase of temperature and the respective contour plot, for the Yb/Eu doped Y_2_O_3_ nanostructures produced using all three acids. It can be observed that for all materials, the complete phase shift to crystalline Y_2_O_3_ occurs at 700 °C, without expressive changes up to 1000 °C.

Figure 4 shows the XRD diffractograms of Yb/Eu doped Y_2_O_3_ nanostructures, prepared with different types of acids, and annealed at 700 °C, for 6 h. For comparison, the XRD simulated Y_2_O_3_ powder pattern is also shown. It is possible to observe that regardless the used acid, after calcination, it was obtained crystalline Y_2_O_3_ nanostructures, having a cubic type structure, with the main reflections being (222), (400), (440) and (622), which is in accordance to the literature [23,42]. No peaks shift, or other impurity phases were detected, indicating that a high purity Yb/Eu doped Y_2_O_3_ nanostructures were obtained by annealing at 700 °C for 6 h. 

The crystallite size, D, was estimated using the Scherrer’s equation and the most intense peak, corresponding to (222) plane [37]:(7)$$D=\frac{0.9\;\lambda }{Bcos\theta }$$
where λ is the X-ray wavelength (corresponding to Cu K_α_ radiation—1.540598 Å), B is the full width at half-maximum (FWHM) in radian and θ is the Bragg diffraction angle degrees. It was obtained a crystallite size of 16.12 nm, 21.88 and 19.86 nm for synthesis with acetic, hydrochloric and nitric acids, respectively.

In order to understand the different transformations that occur during Yb/Eu doped Y_2_O_3_ calcination, differential scanning calorimetric measurements were carried out for all materials. The results are shown in Figure 5. The microwave synthesized materials, which consist in yttrium oxycarbonates were analyzed and it is possible to observe a small mass loss before 110–200 °C, that corresponds to a possible vaporization of the sorbet water. The second weight loss step, below 400–510 °C can be attributed to a dehydration of the yttrium oxycarbotates. The complete conversion into Y_2_O_3_ occurs at temperatures between 400 and 800 °C corresponding to a mass loss step of 20% for hydrochloric and nitric acid and only 12% for acetic acid. This mass loss is accompanied by an endothermic peak centered at 611–650 °C. Above this temperature, no other weight loss steps or peaks are observed, indicating that the complete conversion of the yttrium oxycarbonate into Y_2_O_3_ has occurred. These results are in agreement to what is described in literature [45,48] and to what was observed on *in situ* XRD results (Figure 3). These results justify the selected calcination temperature used in the present study, since above 700 °C, no other phase transformation could be detected. 

SEM and STEM analyses were carried out for all the materials produced. Figure 6 shows SEM images of the materials after microwave synthesis and before and after calcination. As can be seen, the shape of the nanostructures is maintained after calcination. Nevertheless, after heat exposure, the shape of such structures is better defined, especially for the nitric-based structures. When comparing the three acids used, it is evident that the acetic acid resulted in thin nanosheets, while both hydrochloric and nitric resulted in perfect sphere-like structures (Figure 6 and Figure 7). It is also clear that after calcination, it is observed a reduced diameter that can be explained by sintering, where small primary single crystals diffuse across the boundaries and coalescence to form a larger one. The total volume decreased because a densely packed with the elimination of pores was formed.

The nanosheets had an average size of ~2.5 μm and thickness of ~50 nm, while the size of the spheres varied with the acid used. For the hydrochloric acid, the structures had 192 ± 25 nm, and for the nitric, it has been measured an average size of 216 ± 37 nm. Nevertheless, both hydrochloric and nitric-based materials revealed heterogeneities in sphere sizes (this can be observed by the presence of smaller spheres). From Figure 7e,f, it can be observed that the spheres are composed by smaller primary grains, joined to form a compact/granular spherical structure [49]. These smaller grains may justify the crystallite sizes measured by XRD. 

EDS measurements were also carried out on both nanostructures formed (see Figure 8). Both nanosheets and spheres showed the presence of Y, Yb, Eu, and O well distributed along the structures. This indicates that the doped Yb^3+^ and Eu^3+^ were successfully incorporated into the Y_2_O_3_ host matrix. Moreover, these results are in agreement to XRD, where no second phases were detected.

The possible reasons for these morphological differences in the nanostructures studied are expected to be due to a group of distinct factors. Indeed, the shape of Y_2_O_3_ nanostructures can be greatly influenced by factors such as intrinsic crystal/cluster structure configuration, precursors, catalysts, surfactants and also by synthesis temperature and time [50]. However, it is believed that the catalyst is responsible for the final nanostructures’ morphology. The decomposition of the catalyst into ammonia and carbonates ions (OH^−^ and CO_3_^2−^ ions) originates a homogeneous nucleation of Y_2_O_3_ nanostructures occurring a spontaneous aggregation process in order to minimize the surface energy, forming sphere-shaped nanostructures [50,51]. These spherical nanostructures are an intermediate step in the morphological evolution into nanosheets structures. By increasing the amount of catalyst or by increasing the synthesis time, the dissolution–recrystallization equilibrium will cause the dissolution of smaller particles and the anisotropically grow of larger particles (into sheets) during the Ostwald ripening period [50]. Liu et al. [52] synthesized Y_2_O_3_:Eu^3+^ nanospheres and by increasing synthesis time while keeping constant the catalyst concentration, they obtained nanoparticles with irregular shape. Khachatourian et al. [51] also studied the effect of catalyst on the morphology of Y_2_O_3_ nanoparticles. They observed that by increasing the catalyst into a critical point, the diameter nanosphere would decrease (a higher degree of supersaturation would lead into a burst formation of nuclei due to the presence of a higher quantity of OH^−^ and CO_3_^2−^ ions, originating a decrease on particle size). Above that point it would result in larger particles [51]. A higher concentration of catalyst will act both as hydrogen-bond donors and acceptors (through C=O groups), resulting in self-association into two-dimensional layered frameworks (sheets nanostructures) [50,53]. 

By using carboxylic acid (such as acetic acid), it is possible to synthesize a series of nanocrystals connected orthogonally by hydrogen-bonded NH chains and carboxylic acid dimers, and by this way design two-dimensional layered networks [53]. In the case of the other acids tested, hydrochloric and nitric acids resulted in sphere-like structures—but with differences in size—which can be expected since the dissociation reaction of primary oxides should be different regarding the acid used.

### 3.2. Optical Characterizations

The optical band gaps of Yb/Eu doped Y_2_O_3_ nanostructures were evaluated from diffuse reflectance data and through the Tauc relation using Equation (8):(8)$${\left(\alpha h\nu \right)}^{m}=A\left(h\nu -Eg\right)$$
where *α* is the absorption coefficient, *h**ν* is the energy of photons impinging on the material, *A* is a constant of proportionality function of the matrix density of states and *E_g_* is the optical band gap [54]. For such materials, it has been reported that they exhibit a direct allowed transition [55]. Few studies reported the optical band gaps for analogous materials, nevertheless Halappa et al. [55] described Eu^3+^ activated Y_2_O_3_ red nanophosphors with a band gap of 5.51 eV. In another study, Shivaramu et al. [56] demonstrated a band gap value of 5.7 eV also to Y_2_O_3_:Eu^3+^ nanophosphors. In the present study, the band gap values calculated were 4.56 eV, 4.36 eV and 4.53 eV for the materials produced with acetic, hydrochloric and nitric acids, respectively (Figure 9). The values are expressively low, when compared to the literature; however, it is known that adding dopants into the Y_2_O_3_ lattice causes energy levels in the energy gap between the conduction and valence bands (energy levels of donors or acceptors), which will lead to the decrease of the optical band gap [57]. In fact, Cabello-Guzmán et. al. [57] reported the progressive decrease of optical band gap by having two (Y_2_O_3_:Er –5.02 eV) or three dopants (Y_2_O_3_:Er-Yb –4.78 eV) in the host matrix. Moreover, it is expected that material defects and oxygen vacancies can also form discrete energy levels, influencing so the nanostructure’s band gaps. The band gap differences observed between the materials can also be justified by the size and structure disparities observed. 

No direct relation between the nanostructure sizes measured by SEM/STEM and the band gaps values can be stated, however, considering the XRD results (Figure 4), a trend can be inferred, since the lower crystalline size material achieved the highest band gap value (acetate-based material), while the hydrochloric material revealed the lowest band gap value and displayed the highest crystallite size. It is normally accepted that the band gap is strongly dependent on crystallite size [58,59,60], and that the decrease in the band gap can be related to the increase in the grain size [58]. Moreover, other factors such as the degree of compactness and densification can also contribute to the final band gap value [61].

Figure 10 shows the emission spectra of the Yb^3+^:Eu^3+^ doped Y_2_O_3_ nanostructures prepared with different acids and a schematic of the relevant luminescence transitions between the ^5^D_0_ and the ^7^F*_J_* states [44,62]. The produced materials exhibited a strong red emission under excitation with a 976 nm laser. Nevertheless, it is known that the Eu^3+^ ions cannot be excited with an excitation source around 980 nm due to a considerable mismatch of energy levels in Eu^3+^ ions. However, when co-doped with Yb^3+^, the upconversion emission occurs, indicating that the Eu^3+^ ions are excited with the Yb^3+^ ions [2,63].

When exposed to a 976 nm source, pairs of Yb^3+^ ions are excited from its ^2^F_7/2_ ground state to the ^2^F_5/2_. Then, their energy is transferred cooperatively in a way that one of them (acceptor) after gaining energy from the donor, occupies the virtual state (V) (i.e., ^2^F_5/2_ + ^2^F_5/2_ → 2^2^F_5/2_). At this point, the excited Yb^3+^ ions from the virtual state transfer their excitation energy into the ground state of Eu^3+^ ions. Therefore, the Eu^3+^ ion is assumed to transit to the ^5^D_2_ multiplet term or to the ^5^D_1_ multiplet, followed by a nonradiative decay of the ^5^D_0_, and thus by emission [2,63].

The emission spectra observed in Figure 10 revealed radiative emissions at ~586 nm, 611 nm, and 628 nm, for all the materials produced. The emission bands can be assigned to the ^5^D_0_ → ^7^F*_J_* (*J* = 0, 1, 2) transitions of Eu^3+^ [63]. The strongest peak at 611 nm is originated from ^5^D_0_ to ^7^F_2_ transition. When comparing between the material’s emission intensities, it can be observed that the emission bands are sharpen for the hydrochloric and nitric acid-based materials (the maximum values for the ^5^D_0_–^7^F_2_ transitions were 3050 and 2840, for hydrochloric and nitric acid-based materials, respectively). The lower emission observed for the acetic-based material (maximum value of 1650 for the ^5^D_0_–^7^F_2_ transition) can be associated to a size effect, since the luminescent intensity is known to be inversely proportional to the particle size, and thus the larger sized nanosheets are expected to have lower luminescence. Nevertheless, several aspects may influence the relative emission intensities of all materials, including absorption cross-section, competition between radiative cooperative emission, energy transfer to Eu^3+^ and energy migration to traps [63], besides a mesoscopic effect, associated to the shape and configuration of the nanostructures produced.

## 4. Conclusions

The fast microwave syntheses (15 min) of Yb/Eu doped Y_2_O_3_ nanostructures was reported. Microwave syntheses revealed itself to be an efficient way for successfully doping the Y_2_O_3_ host matrix after dissociating the primary oxides with different acids. In fact, this study revealed the key role of the acids used regarding the final structure and optical behaviour of the nanostructures produced. SEM and STEM revealed that the acetic acid resulted in nanosheets, while both hydrochloric and nitric formed nanospheres with heterogeneities in size. The study indicated the calcination temperature (700 °C) at which the conversion into Y_2_O_3_ is complete for such nanostructures. This temperature range is expected to cause an impact on production and energy costs of such nanostructures—in fact the calcination temperature is substantially lower than what is usually reported in literature. The optical properties of the produced materials were discussed, and the photoluminescence experiments showed a red Eu^3+^ emission when exited to a 976 nm source. Moreover, the sphere-like structures revealed enhanced luminescence, when compared to the nanosheets. The lower emission of the nanosheets was associated to a size effect, since acetic acid originated larger structures. The present study opens up to the possibility of these materials to be used as infrared to visible upconverters, and their potential to be integrated in several opto-electronic devices. For further studies, it is imperative to control the particle size and the size distribution of the developed nanostructures, which can be achieved by altering the catalyst or its concentration, but also with further investigation and adjustments on the synthesis parameters.

## Figures and Tables

**Figure 1 nanomaterials-09-00234-f001:**
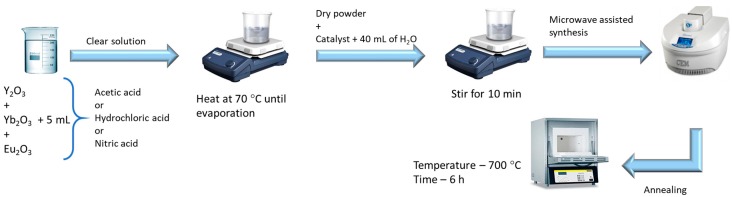
Schematic of Yb/Eu doped Y_2_O_3_ nanostructures produced by a hydrothermal process under microwave irradiation.

**Figure 2 nanomaterials-09-00234-f002:**
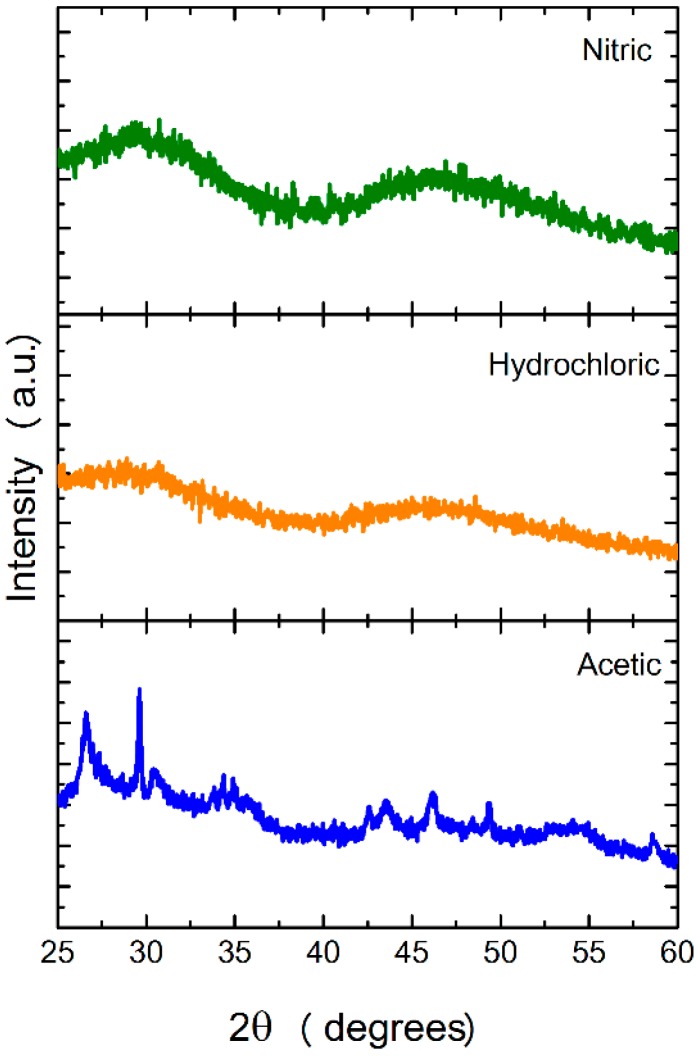
X-Ray diffraction (XRD) diffractograms of nanostructures after microwave synthesis and before any calcination, for the three sets of reactions defined above.

**Figure 3 nanomaterials-09-00234-f003:**
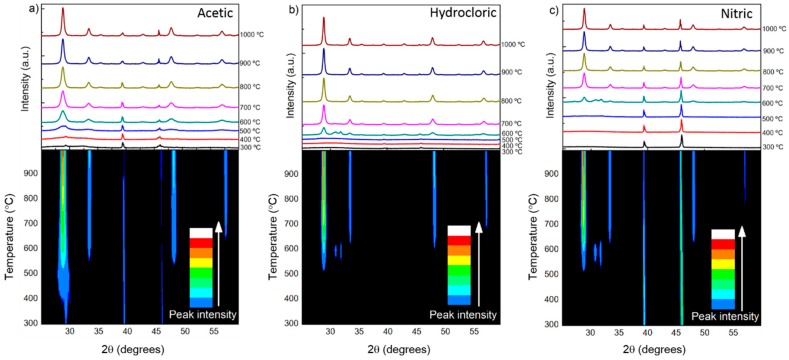
*In situ* XRD diffractograms as a function of temperature (on top) and the respective contour plot (on bottom) of Yb/Eu doped Y_2_O_3_ nanostructures after microwave synthesis, when using (**a**) acetic, (**b**) hydrochloric and (**c**) nitric acids.

**Figure 4 nanomaterials-09-00234-f004:**
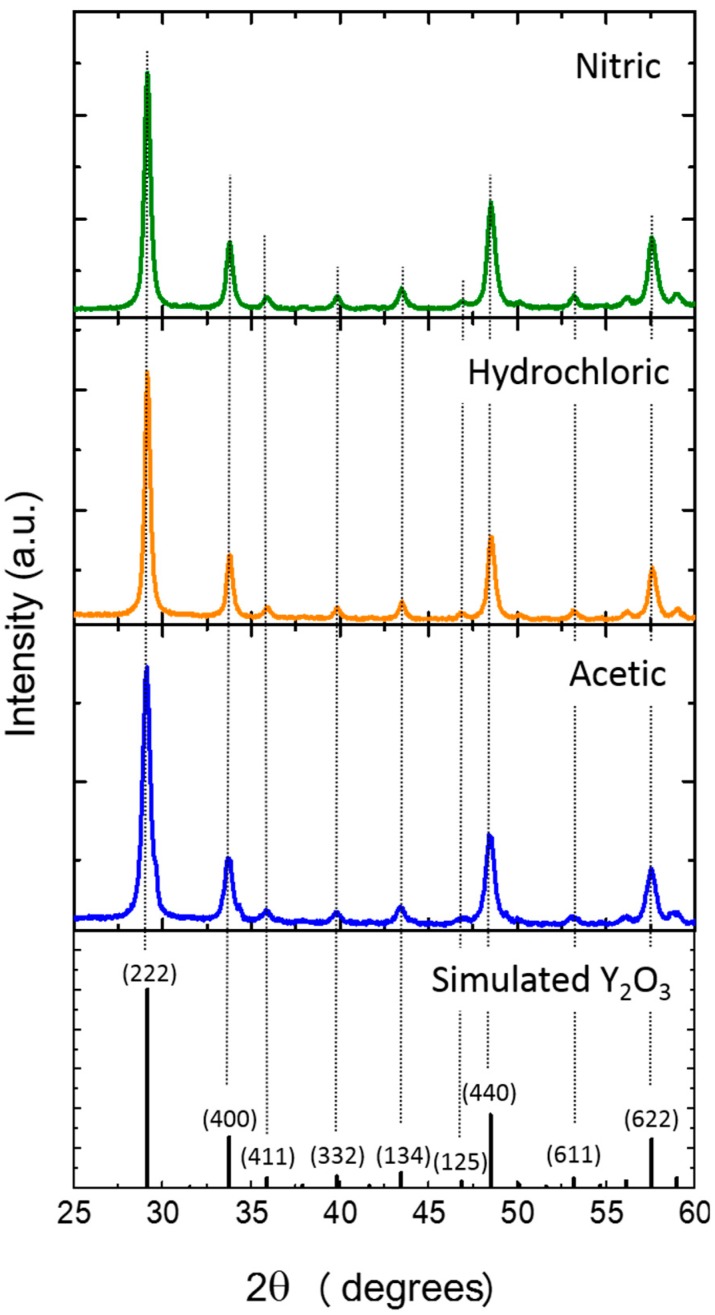
XRD diffractograms of Yb/Eu doped Y_2_O_3_ nanostructures after calcination at 700 °C. The simulated Y_2_O_3_ powder pattern is also shown for comparison.

**Figure 5 nanomaterials-09-00234-f005:**
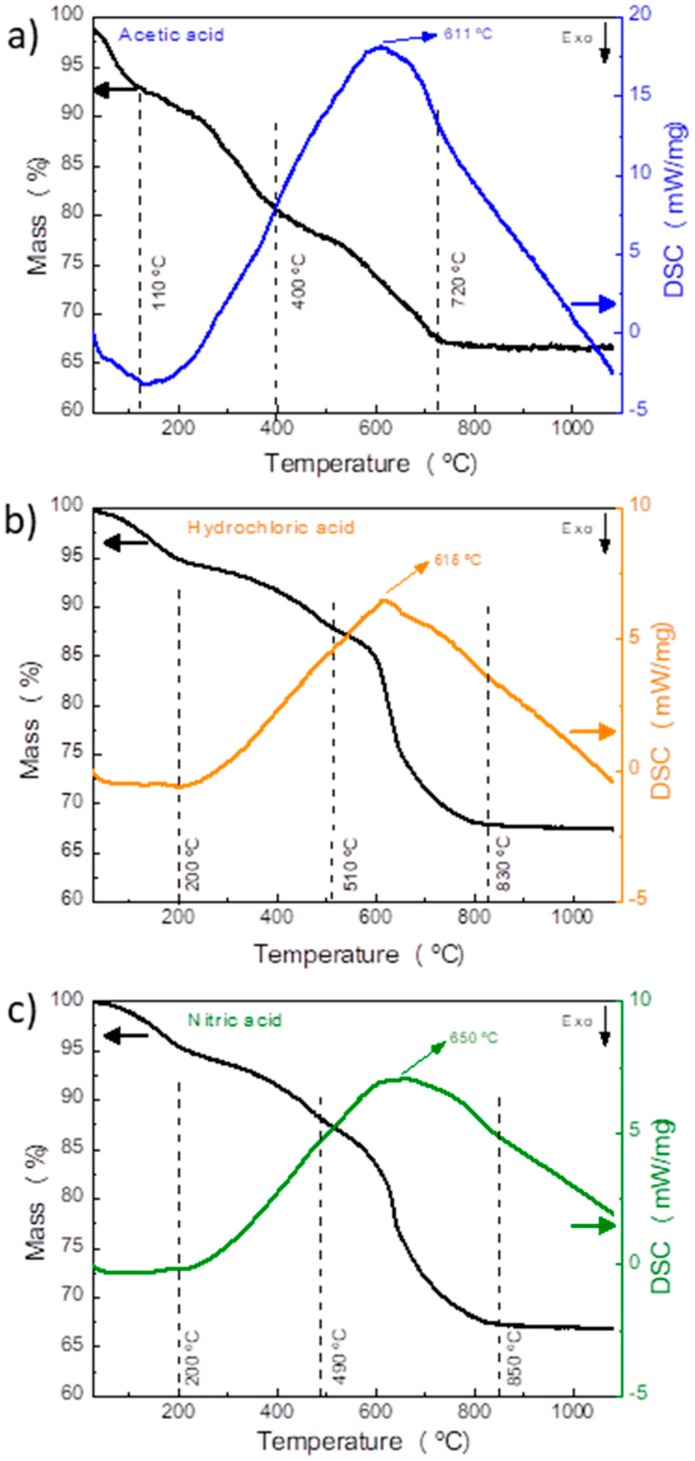
Thermogravimetric analysis and differential scanning calorimetry (TGA/DSC) curves of the as-synthesized nanostructures before calcination produced with (**a**) acetic acid, (**b**) hydrochloric acid and (**c**) nitric acid.

**Figure 6 nanomaterials-09-00234-f006:**
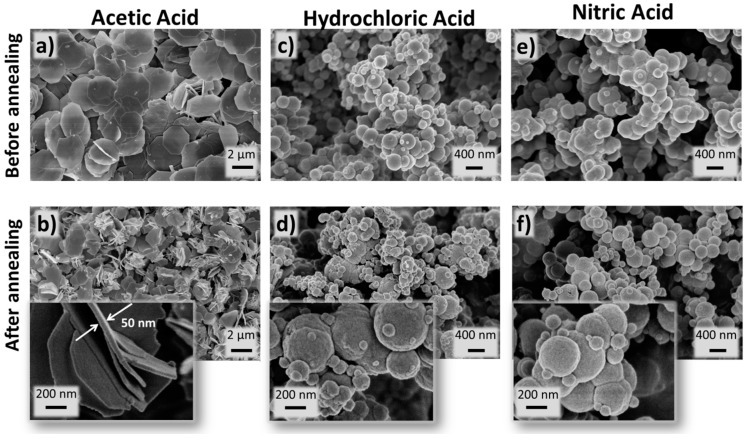
Scanning electron microscopy (SEM) images of Yb/Eu doped Y_2_O_3_ nanostructures produced by hydrothermal method assisted by microwave radiation, and using (**a**,**b**) acetic acid, (**c**,**d**) hydrochloric acid and (**e**,**f**) nitric acid, before and after annealing at 700 °C, respectively. The insets magnify the structures produced after calcination.

**Figure 7 nanomaterials-09-00234-f007:**
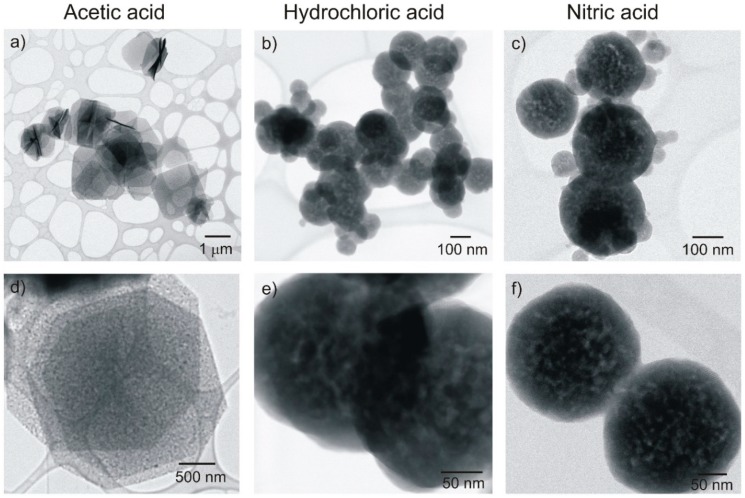
Scanning transmission electron microscopy (STEM) images of Y_2_O_3_:Yb:Eu nanostructures produced by hydrothermal method assisted by microwave irradiation, and using (**a**,**d**) acetic acid, (**b**,**e**) hydrochloric acid, and (**c**,**f**) nitric acid after calcination at 700 °C.

**Figure 8 nanomaterials-09-00234-f008:**
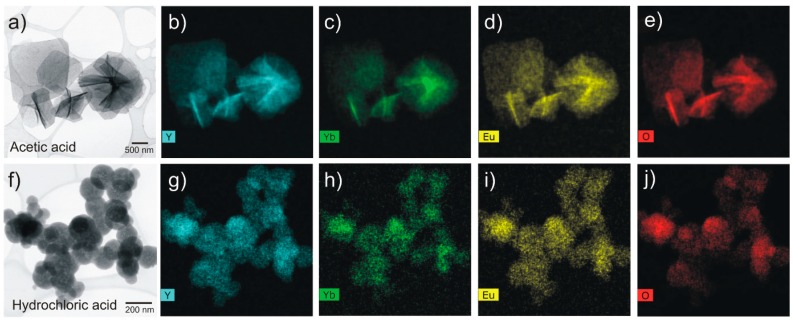
STEM images (**a**,**f**) and energy dispersive X-Ray spectroscopy (EDS) analyses of the Yb/Eu doped Y_2_O_3_ nanostructures after microwave synthesis and calcination at 700 °C. The corresponding EDS maps for Y (**b**,**g**), Yb (**c**,**h**), Eu (**d**,**i**), and O (**e**,**j**) are presented. The two types of structures have been analyzed.

**Figure 9 nanomaterials-09-00234-f009:**
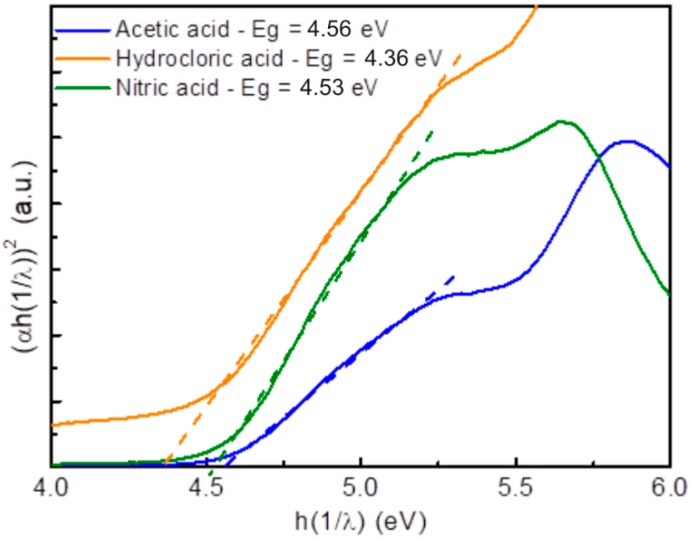
Tauc plot for the Yb/Eu doped Y_2_O_3_ nanostructures produced with acetic, hydrochloric and nitric acids, respectively.

**Figure 10 nanomaterials-09-00234-f010:**
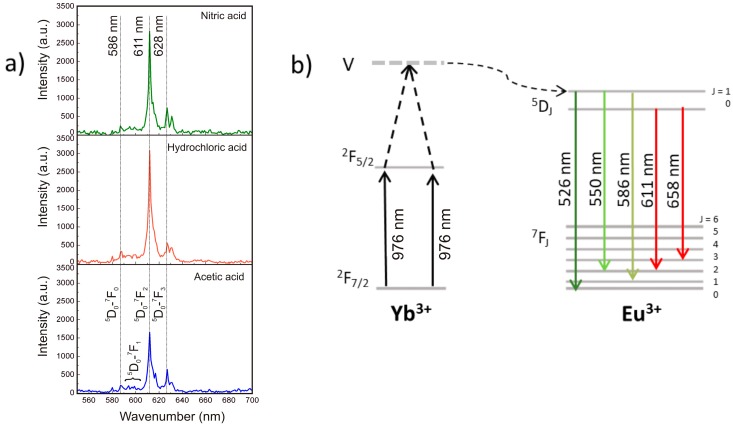
(**a**) Emission spectra of the Yb/Eu doped Y_2_O_3_ nanostructures obtained at 976 nm, and (**b**) Energy level diagram for Yb^3+^ and Eu^3+^ together with the relevant luminescence transitions between the ^5^D_0_ and the ^7^F_J_ states.

**Table 1 nanomaterials-09-00234-t001:** Transitions of lanthanide elements [3,4,5,6].

Element	Color of the Visible Luminescence	Transition	Intensity
Ce, cerium	UV	5d → ^2^F_5/2_ (300–450 nm)	n.a.
Pr, praseodymium	Red	^1^D_2_ → ^4^H_4_ (600 nm) ^3^P_0_ → ^3^F_2_ (700 nm)	Weak
Nd, neodymium	Infra-red	^4^F_3/2_ → ^4^I_9/2_ (900 nm)	n.a.
Sm, samarium	Orange–red	^4^G_5/2_ → ^6^H_7/2_ (600 nm)	Medium
Eu, europium	Red–orange	^5^D_0_ → ^7^F_4_ (720 nm) ^5^D_0_ → ^7^F_3_ (650 nm) ^5^D_0_ → ^7^F_2_ (615 nm) ^5^D_0_ → ^7^F_1_ (590 nm) ^5^D_0_ → ^7^F_0_ (580 nm)	Strong
Gd, gadolinium	UV	^6^P_7/2_ → ^8^S_7/2_ (310 nm)	Strong
Td, terbium	Green–orange	^5^D_4_ → ^7^F_5_ (540 nm) ^5^D_4_ → ^7^F_4_ (580 nm) ^5^D_4_ → ^7^F_3_ (620 nm) ^5^D_4_ → ^7^F_2_ (650 nm) ^5^D_4_ → ^7^F_1_ (660 nm)	Strong
Dy, dysprosium	Yellow	^4^F_9/2_ → ^6^H_13/2_ (570 nm) ^4^I_15/2_ → ^6^H_13/2_ (540 nm)	Medium
Ho, holmium	Red	^5^F_5_ → ^5^I_8_ (650 nm)	Weak
Er, erbium	Green	^4^S_3/2_ → ^4^I_15/2_ (545 nm) ^4^F_9/2_ → ^4^I_15/2_ (660 nm)	Weak
Tm, thulium	Blue	^1^G_4_ → ^3^H_6_ (470 nm)	Weak
Yb, ytterbium	Infra-red	^2^F_5/2_ → ^7^F_7/2_ (980 nm)	n.a.

n.a.—not available.

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
