# Peer review of "Tailoring Upconversion and Morphology of Yb/Eu Doped Y2O3 Nanostructures by Acid Composition Mediation"

_nanomaterials, 2019, doi:10.3390/nano9020234_

Reviewer 1 Report

In the manuscript entitled “Tailoring upconversion and morphology of Yb/Eu doped Y2O3 nanostructures by acid composition mediation” the authors have successfully grown different type of doped Y2O3 nanostructure by controlling the various acetic environment. The manuscript contains more systematic and interesting studies. However, the author should address some of the comments relate to the manuscript.

1.      The XRD data for the composite structure is very interesting and the doping level of YbO3 into the Y2O3 is close to 10% (0.05 M of Y2O3, 0.01 M of YbO3 and 0.0075 M of Eu2O3). However, the authors could not see any impurities, structural distortion or any type of identification peak of the composite structure rather than the Y2O3 itself in the XRD result. There may be any possibilities of XRD peak if the analyses the XRD data can be done using logarithmic scale.

2.      The crystallite size using Scherrer’s equation for the Yb/Eu doped Y2O3 nanostructures are 16.12 nm, 21.88 and 19.86 nm for synthesis with acetic, hydrochloric and nitric acids, 207 respectively. However, the SEM/STEM results show that the nitric acids environment grown Yb/Eu doped Y2O3 is smaller in size as compare to the others. The author should explain the results more clearly for the better understanding of the readers.

3.      Band gap is always strongly related to the particle size. Higher the particle size lower the band gap. However, the present study deviate the behavior and the band gap values calculated from Tauc plot are 4.56 eV, 4.36 eV and 4.53 eV for the materials  produced with acetic, hydrochloric and nitric acids, respectively, whereas the acetic environment grown particle is larger in size but showing higher band gap. The author should explain this behavior in the revised manuscript.      

Author Response

Caparica, January 23rd, 2019

Dear Sirs,

The authors acknowledge the pertinence of the comments and have revised the manuscript by taking them into account. Following the reviewer’s comments order, here are our answers:

Reviewer 1: In the manuscript entitled “Tailoring upconversion and morphology of Yb/Eu doped Y2O3 nanostructures by acid composition mediation” the authors have successfully grown different type of doped Y2O3 nanostructure by controlling the various acetic environment. The manuscript contains more systematic and interesting studies. However, the author should address some of the comments relate to the manuscript.

1.      The XRD data for the composite structure is very interesting and the doping level of YbO3 into the Y2O3 is close to 10% (0.05 M of Y2O3, 0.01 M of YbO3 and 0.0075 M of Eu2O3). However, the authors could not see any impurities, structural distortion or any type of identification peak of the composite structure rather than the Y2O3 itself in the XRD result. There may be any possibilities of XRD peak if the analyses the XRD data can be done using logarithmic scale.

·The authors acknowledge the reviewer’s comment, however no composite structure has been formed. Here we present the logarithmic scale graph, and as presented in the main manuscript, a single phase is observed, i.e. Y2O3.

             Figure 1 – XRD diffractograms represented in logarithmic scale for all the materials produced after calcination at 700 ºC.

2.      The crystallite size using Scherrer’s equation for the Yb/Eu doped Y2O3 nanostructures are 16.12 nm, 21.88 and 19.86 nm for synthesis with acetic, hydrochloric and nitric acids, respectively. However, the SEM/STEM results show that the nitric acids environment grown Yb/Eu doped Y2O3 is smaller in size as compare to the others. The author should explain the results more clearly for the better understanding of the readers.

· The process for producing such structures is explained in Pages 11 and 12, suggesting that the formation of sphere-like structures is an intermediate step for the evolution into nanosheets. Moreover, it has been also explained that the catalyst used play an important role on the size of nanocrystal, and by using carboxylic acid, the structures are prone to grow following a 2-dimentional network. In the new version of the manuscript, a sentence stating the differences in size for hydrochloric and nitric acid-based materials has been included: “In the case of the other acids tested, hydrochloric and nitric acids resulted in sphere-like structures, however with differences in size, which can be expected since the dissociation reaction of primary oxides should be different regarding the acid used” (page 12).

3.      Band gap is always strongly related to the particle size. Higher the particle size lower the band gap. However, the present study deviate the behavior and the band gap values calculated from Tauc plot are 4.56 eV, 4.36 eV and 4.53 eV for the materials  produced with acetic, hydrochloric and nitric acids, respectively, whereas the acetic environment grown particle is larger in size but showing higher band gap. The author should explain this behavior in the revised manuscript.       

·The author’s acknowledge the reviewer’s comments and a paragraph has been included in the revised manuscript ”No direct relation between the nanostructure sizes measured by SEM/STEM and the band gaps values can be stated, however considering the XRD results (Figure 4), a trend can be inferred, since the lower crystalline size material, achieved the highest band gap value (acetate-based material), while the hydrochloric material, revealed the lowest band gap value and displayed the highest crystallite size. It is normally accepted that the band gap is strongly dependent on crystallite size [58–60], and that the decrease in the band gap can be related to the increase in the grain size [58]. Moreover, other factors such as the degree of compactness and densification can also contribute to the final band gap value [61]” (page 12).

Yours faithfully,

Daniela Nunes

Invited Assistant Professor
CENIMAT/I3N, Faculdade de Ciências e Tecnologia - Universidade Nova de Lisboa

CEMOP-UNINOVA

2829-516 Caparica, Portugal

http://www.cenimat.fct.unlNaN/

daniela.gomes@fct.unlNaN

Reviewer 2 Report

The authors present an original way to produce upconversion nanostructures through first the dissociation of primary oxide materials before a hydrothermal method to obtain nanostructures.

The acid used for oxide dissolution is a key factor on the final particles size and morphology. The obtained materials are deeply and well characterized.   

Few comments:

1)    Even if methods are briefly described within the discussion, I think that authors should provide complete description within a dedicated section. What is the concentration of each acid? Optical properties are measured within solution or powder? Etc.

2)    The authors provide a wide background on the different synthesis of Y2O3 based upconversion nanosystems. How do the authors place their synthesis in relation to these syntheses described in the literature? Advantages/disadvantages

3)    Authors are concluding that they present a fast MW assisted method. If this step is fast, there are also dissolution and calcination process, which are time consuming. Did the authors test their method using metal salts to avoid the dissolution step?

4)    The obtained particles are polydispersed and too large for biomedical applications. Authors could add in conclusion, perspectives for a better size control.

Author Response

Caparica, January 23rd, 2019

Dear Sirs,

The authors acknowledge the pertinence of the comments and have revised the manuscript by taking them into account. Following the reviewer’s comments order, here are our answers:

Reviewer 2: The authors present an original way to produce upconversion nanostructures through first the dissociation of primary oxide materials before a hydrothermal method to obtain nanostructures.

The acid used for oxide dissolution is a key factor on the final particles size and morphology. The obtained materials are deeply and well characterized.   

Few comments:

1)    Even if methods are briefly described within the discussion, I think that authors should provide complete description within a dedicated section. What is the concentration of each acid? Optical properties are measured within solution or powder? Etc.

·    The authors acknowledge the reviewer’s comments, however the Materials section (2.1 Materials) describes in detail the materials syntheses. Nevertheless, a sentence has been included saying that all the characterization techniques analysed the materials in the powder form.

2)    The authors provide a wide background on the different synthesis of Y2O3 based upconversion nanosystems. How do the authors place their synthesis in relation to these syntheses described in the literature? Advantages/disadvantages

·     The synthesis of doped Y2O3 is a hot scientific topic nowadays, with the application of such materials in several opto-electronic devices. However, in many published studies, the synthesis of such materials required time and energy consuming techniques. Nevertheless, as described in the manuscript, microwave synthesis has the advantages of being fast, reproducible and cost-efficient, which are also advantages of our Y2O3 synthesis (15 min synthesis time). In terms of using acids for oxide dissociation, it is a key step, but in some sense can be considered the main bottleneck of the whole approach, requiring specialized personal and certain security concerns in the synthesis laboratory. Turning to the calcination process, this had an increment in terms of temperature range and when compared to studies published. It has been discussed in detail the minimal temperature required for Y2O3 phase transformation.

3)    Authors are concluding that they present a fast MW assisted method. If this step is fast, there are also dissolution and calcination process, which are time consuming. Did the authors test their method using metal salts to avoid the dissolution step?

·     The authors acknowledge the reviewer’s comments, nevertheless, the synthesis method presented in this report is fast and reproducible, only taking 15 minutes. The calcination process is the most time-consuming step, but it is mandatory, regardless the method used. The dissolution step could be avoided if other precursors were used, but the purpose of this work was to study the influence of the acids used for dissociation of primary oxides and their impact on structural, morphological and optical properties of the resulting materials.

4)    The obtained particles are polydispersed and too large for biomedical applications. Authors could add in conclusion, perspectives for a better size control.

·      The authors acknowledge the reviewer’s comments and a paragraph has been included in the revised manuscript, in the conclusions section: “For further studies, it is imperative to control the particle size and the size distribution of the developed nanostructures, that can be possible by altering the catalyst or its concentration, but also with further investigation and adjustments on the synthesis parameters.”

Yours faithfully,

Daniela Nunes

Invited Assistant Professor
CENIMAT/I3N, Faculdade de Ciências e Tecnologia - Universidade Nova de Lisboa

CEMOP-UNINOVA

2829-516 Caparica, Portugal

http://www.cenimat.fct.unlNaN/

daniela.gomes@fct.unlNaN
